# Culture of *Mycobacterium smegmatis* in Different Carbon Sources to Induce In Vitro Cholesterol Consumption Leads to Alterations in the Host Cells after Infection: A Macrophage Proteomics Analysis

**DOI:** 10.3390/pathogens10060662

**Published:** 2021-05-28

**Authors:** Jaqueline Batista de Lima, Lana Patricia da Silva Fonseca, Luciana Pereira Xavier, Barbarella de Matos Macchi, Juliana Silva Cassoli, Edilene Oliveira da Silva, Rafael Borges da Silva Valadares, José Luiz Martins do Nascimento, Agenor Valadares Santos, Chubert Bernardo Castro de Sena

**Affiliations:** 1Laboratory of Structural Biology, Institute of Biological Sciences, Federal University of Pará, Belém 66075-110, PA, Brazil; jaqueline.lima@icb.ufpa.br (J.B.d.L.); edilene@ufpa.br (E.O.d.S.); 2Laboratory of Biotechnology of Enzymes and Biotransformations, Institute of Biological Sciences, Federal University of Pará, Belém 66075-110, PA, Brazil; lpxavier@ufpa.br (L.P.X.); avsantos@ufpa.br (A.V.S.); 3Instituto Tecnológico Vale, Belém 66055-090, PA, Brazil; lanafonseca@ufrn.edu.br (L.P.d.S.F.); Rafael.borges.valadares@itv.org (R.B.d.S.V.); 4Laboratory of Molecular and Cellular Neurochemistry, Institute of Biological Sciences, Federal University of Pará, Belém 66075-110, PA, Brazil; barbarella@ufpa.br (B.d.M.M.); jlmn@ufpa.br (J.L.M.d.N.); 5National Institute of Science and Technology in Neuroimmunomodulation (INCT-NIM), Rio de Janeiro 21040-900, RJ, Brazil; 6Institute of Biological Sciences, Federal University of Pará, Belém 66075-110, PA, Brazil; jscassoli@ufpa.br; 7National Institute of Science and Technology in Structural Biology and Bioimaging, Rio de Janeiro 21941-901, RJ, Brazil

**Keywords:** *Mycobacterium*, proteomics, cholesterol, infection, macrophages

## Abstract

During tuberculosis, *Mycobacterium* uses host macrophage cholesterol as a carbon and energy source. To mimic these conditions, *Mycobacterium smegmatis* can be cultured in minimal medium (MM) to induce cholesterol consumption in vitro. During cultivation, *M. smegmatis* consumes MM cholesterol and changes the accumulation of cell wall compounds, such as PIMs, LM, and LAM, which plays an important role in its pathogenicity. These changes lead to cell surface hydrophobicity modifications and H_2_O_2_ susceptibility. Furthermore, when *M. smegmatis* infects J774A.1 macrophages, it induces granuloma-like structure formation. The present study aims to assess macrophage molecular disturbances caused by *M. smegmatis* after cholesterol consumption, using proteomics analyses. Proteins that showed changes in expression levels were analyzed in silico using OmicsBox and String analysis to investigate the canonical pathways and functional networks involved in infection. Our results demonstrate that, after cholesterol consumption, *M. smegmatis* can induce deregulation of protein expression in macrophages. Many of these proteins are related to cytoskeleton remodeling, immune response, the ubiquitination pathway, mRNA processing, and immunometabolism. The identification of these proteins sheds light on the biochemical pathways involved in the mechanisms of action of mycobacteria infection, and may suggest novel protein targets for the development of new and improved treatments.

## 1. Introduction

Tuberculosis (TB) is a communicable disease that is a major cause of ill health worldwide. A quarter of the global population is infected with the causative agent, *Mycobacterium tuberculosis* (*M. tuberculosis*), and thus at risk of developing TB disease. It is estimated that 1.2 million people died of TB in 2019 [1]. TB is transmitted by aerosols in droplets containing the bacteria from an infected individual to a healthy individual. The improvement of knowledge regarding the host–pathogen interaction is one manner of overcoming obstacles to control TB. 

*M. tuberculosis* is a successful human pathogen that is able to grow and replicate within a host and its success as an infectious agent is due, in large, to its ability to persist in the macrophages [2]. The pathogenesis of tuberculosis is controlled by a complex interaction between the host immune system and survival strategies developed by *M. tuberculosis* [2].

Alveolar macrophages are the most important immune cell defense against *M. tuberculosis* and, in addition to inducing the innate immune response, they also play central roles in TB control. These cells recognize and phagocyte pathogens, such as *M. tuberculosis*, which resides in a membrane-bound vacuoles inside the host, the phagosome, which usually matures into phagolysosomes. Other mechanisms employed to kill *M. tuberculosis* include the production of reactive oxygen and nitrogen intermediates, cytokine generation, up regulation of the expression of antimicrobial peptides, classical killing pathways such as autophagy and apoptosis, lipid mediator release, and sequestration of cofactors. Disruption of any of these macrophage functions affects the immune response [3,4].

Although macrophages are thought to be an effective barrier against pathogens, *M. tuberculosis* has evolved mechanisms to evade the host immune response, thereby creating a favorable environment for intracellular replication via the exploitation of cell wall components such as phosphatidylinositol mannosides (PIMs), lipomannan (LM), lipoarabinomannan (LAM), trehalose dimycolate (TDM), glycopeptidolipid (GPL), and mycolic acids. All of these compounds are key modulators of host immune processes and leads to the inhibition of phagosome-lysosome fusion and phagosome acidification, the modulation of macrophage membrane properties and the host cell cytoskeleton, as well as host cell signaling inhibition, programmed death mechanisms, downregulation of host gene expression, and formation of granuloma [3,5,6,7,8,9].

Many of these processes are modified by *M. tuberculosis*, which causes inhibition, activation, recruitment, retention, or accumulation of several proteins as a strategy for survival in the hostile environment of the host. Some of these proteins are TACO/coronin-1 which is recruited and retained to block phagosome-lysosome fusion, calcineurin that also contributes to blocking phagosome-lysosome fusion, v-ATPase whose exclusion inhibits phagosome acidification, and many others [10]. In contrast, non-pathogenic mycobacteria, such as *Mycobacterium smegmatis*, reside in phagosomes that are fully mature by fusing with late endocytic compartments, a process that facilitates the killing of these bacteria [11,12]. An emerging topic in TB pathogenesis is the manipulation of host lipid metabolism by *M. tuberculosis*, whereby fatty acids and cholesterol are routed toward intracellular bacilli. Inside the phagosome, *M. tuberculosis* requires host lipids and cholesterol to accumulate lipid droplets and induce foamy macrophages. Host cholesterol has already been shown to be required for optimal growth and persistence of *M. tuberculosis* during infection, where the bacterium uses it as a source of carbon and biosynthetic precursors that are needed to produce virulence-associated molecules [2,13,14,15]. 

In a recent study by our research group, Santos et al. (2019) [16] provided the first evidence that the culture of non-pathogenic *M. smegmatis* in minimal medium (MM) to induce cholesterol consumption leads to several changes in the bacterial cell wall structure. The authors demonstrated that cholesterol availability in MM doubled the amount of intracellular cholesterol, only when glycerol is lacking, showing it to be important for the maintenance of small lipids such as PIMs and phospholipids on the mycobacterial surface. Moreover, *M. smegmatis* also grows in MM, independently of the presence of cholesterol, where it is induced to use mycolic acids to maintain TDM levels, thereby decreasing cell wall mycolate accumulation. In addition to depleting mycolic acid contents in the cell wall, *M. smegmatis* also changes the biosynthesis of LM and LAM, generating unusual molecules. This report demonstrated the importance of mycolic acids and LM and LAM for maintaining the integrity of the *M. smegmatis* cell wall after culturing in minimal growth conditions similar to those occurring in macrophage infection, where bacteria must induce the accumulation of host cholesterol to survive [13,14,17]. In the same work, in response to the new environment and after mycolic acid reduction, GPLs were highly present and were necessary to maintain the integrity of the cell wall and ensure cell survival after culture in MM. All these changes help mycobacteria to modify cell surface hydrophobicity and become resistant to hydrogen peroxide. It is possible that this new MM environment can change the physiology of the mycobacteria to allow the bacilli to become resistant to phagocytosis. In addition, following cell wall modifications, the infection of J774A.1 macrophages with *M. smegmatis*, the host changes its cellular reorganization to form granuloma-like structures. In particular, the high quantities of these structures show that the use of cholesterol by non-pathogenic *M. smegmatis* can potentiate the infection by this mycobacteria [16].

Recently, a number of proteomics researchers have reported the deregulation of several proteins during host–pathogen interactions induced by *M. tuberculosis*, indicating significant changes in biological processes such as apoptosis, blood coagulation, and oxidative phosphorylation [18]. A variety of studies have investigated cells stimulated by infection or mycobacterial bioactive lipids [18,19,20,21,22]. One study showed that the infection of macrophages with mycobacterial cell wall lipids altered the differential expression of proteins involved in immune response, oxidation and reduction, and vesicle transport, as well as other cellular processes [19].

Here, for the first time, using a proteomic study, we show that the infection of macrophages with a fast-growing and non-pathogenic bacterium, *M. smegmatis*, led to several changes in the differential expression of numerous proteins after induction of cholesterol consumption and consequent infection.

## 2. Materials and Methods

### 2.1. Mycobacterial Strain and Culture Conditions

Mycobacterium smegmatis (Trevisan) of the Lehmann and Neumann strain (American Type Culture Collection, ATCC 607; Instituto Nacional de Controle de Qualidade em Saúde, INCQS 00021) were kindly provided by Fundação Oswaldo Cruz-FIOCRUZ, Rio de Janeiro, RJ, Brazil. For each experiment, the strain was grown on Middlebrook 7H10 agar supplemented with 0.5% glycerol, 0.2% glucose, and 14 mM NaCl at 37 °C for 3 days. After this time, cells were cultured until late stationary growth phase in Middlebrook 7H9 broth (BD Biosciences) supplemented with 0.2% glycerol, 0.2% glucose, 0.05% tyloxapol, and 14 mM NaCl at 37 °C for 3 days with agitation at 250 r.p.m [23]. The bacterial culture in the stationary phase of growth was diluted for new 20 mL culture with OD_600_ 0.05 in fresh Middlebrook 7H9 broth or in minimal media (MM) containing compounds like ZnSO_4_ (0.1 mg/L), MgSO_4_·7H_2_O (0.5 g/L), Na_2_HPO_4_ (2.5 g/L), CaCl_2_ (0.5 g/L), KH_2_PO_4_ (1.0 g/L), NH_4_Fe(SO_4_)_2_·12H_2_O (50 mg/L), asparagine (0.5 g/L), vitamin B12 (10 mg/mL), 0.2% tyloxapol, and 0.1% glycerol (in 7H9) or 0.01% cholesterol [14]. The cells were cultured in three groups: (1) 7H9 + Gly (Middlebrook 7H9 broth supplemented with glycerol); (2) MM + Chol (minimal medium supplemented with cholesterol); and (3) MM (only minimal medium). The optical density at 600 nm (A_600_) was used to analyze the bacterial growth [16].

### 2.2. Macrophage Culture and Infection with Mycobacterium Smegmatis

The J774A.1 cells (ATCC TIB-67, Cell Bank of Rio de Janeiro–BCRJ, Rio de Janeiro, RJ, Brazil), a murine macrophage cell line, was cultured in DMEM medium with 10% fetal bovine serum, 100 U/mL penicillin, and 100 μg/mL streptomycin. The cells were maintained at 37 °C in a 5% CO_2_-humidified atmosphere. For infection, macrophages cells (6.6 × 10^5^/flask) were seeded in cell culture flasks and incubated for 2 days until reaching 90% confluence. Bacterial cultures at the early stationary phase were pelleted, washed twice in PBS pH 7.4, bath-sonicated for 15 min to disrupt bacterial clumps, and resuspended in DMEM medium to a final OD_600_ 0.1. Macrophages were infected with *M. smegmatis* (grown in three different bacterial culture mediums) at a multiplicity of infection (MOI) of 100:1. In each experiment, after 1 h infection, extracellular bacteria were dead after addition of 10 µg/mL gentamicin. After 12 h of infection, the cells were washed three times with cold PBS pH 7.4 [11,16]. Macrophages without infection were used as a control group as shown in Figure 1.

### 2.3. Cell Lysis and Sample Preparation

After 12 h of infection, the cells were washed with cold PBS pH 7.4 (three times) and harvested by scraping in PBS. Cells were centrifuged at 1500× *g* for 5 min on ice and resuspended in 6 M urea, 2 M thiourea, and 10 mM dithiothreitol (DTT) for lysis and reduction for 2 h at 37 °C. After incubation, the samples were diluted 10-folds in 20 mM ammonium bicarbonate pH 7.5 and sonicated on ice. For alkylation, 200 mM of iodoacetamide in 20 mM of triethylammonium bicarbonate (to achieve a final concentration of 20 mM iodoacetamide) was added and incubated for 20 min in the dark at room temperature, to reduce and alkylate cysteine residues. Protein concentrations were measured using the Qubit Protein Assay Kit (Invitrogen). The samples were digested with trypsin at 1:50 (*w*:*w*; trypsin:sample) and incubated at 37 °C overnight. The digestion process was stopped by adding 10 μL of 5% trifluoroacetic acid (TFA) [24].

The samples were desalted using a C18 column (SepPack 50 ng Waters), assembling a vacuum system with manifold (Waters) and vacuum pump (Millipore). The column was activated with 100% acetonitrile (ACN), equilibrated with 50% ACN on 0.1% formic acid (FA), and equilibrated again with 1 mL of 0.1% TFA. The sample was previously acidified with 0.4% TFA, loaded onto C18, and the salt removed from the sample with 0.1% TFA. The column was equilibrated again with FA. The sample was eluted with 50% ACN on 0.1% FA, and 80% CAN on 0.1% FA. The samples were concentrated in a SpeedVac evaporator at 25 °C for 24 h, and resuspended in 20 mM ammonium formate, diluted 10 times, in the proportion of 75 μL for each 50 ng of protein in the samples [24].

### 2.4. NanoLC-ESI MS/MS Analysis

Proteomic analysis was performed in a bidimensional nanoUPLC tandem nanoESI-MS/MS platform and the MS/MS fragmentation spectra were acquired in multiplexed data-independent mode (MSE) using a 2D-RP/RP Nano Acquity UPLC System (Waters Corporation, Milford, MA, USA) coupled to a Synapt G2 mass spectrometer (Waters Corporation, Milford, MA, USA). One-dimension reversed-phase (RP) approach was used to fractionate the samples. Peptide samples (0.5 µg) were loaded onto an Acquity UPLC M-Class CSH (C18 packed with changed surface hybrid) Column (100 Å, 1.8 µm, 100 µm × 100 mm; Waters Corporation, Milford, MA, USA) at a flow rate of 2 µL/min. An acetronitrile gradient from 3% to 40% v/v at flow rate 400 nL/min^−1^ for 40 min was used for peptide fractionation directly into a Synapt G2. The mass spectrometer was operated in the resolution mode with an m/z resolving power of about 20,000 FWHM for every measurement. MS and MS/MS data were acquired in positive ion mode in the range of 50–1200 m/z. The low-energy MS mode by applying constant collision energy of 4 eV was used to collected precursor ion information, and the elevated energy scan using a ramped collision energy (19−45 eV) applied to the collision-induced dissociation cell was used to the fragment ion information. The lock mass channel was sampled every 30 sec using 0.1 s scans over the same mass rang. For mass spectrometer calibration was used an MS/MS spectrum of [Glu^1^]-Fibrinopeptide B human (785.8426 *m*/*z*) solution that was delivered through the reference sprayer of the NanoLock Spray source [25].

### 2.5. Data Processing and Database Searches

For identification and quantification of proteins, were used dedicated algorithms and searching against the UniProt Proteomic Database of *Mus musculus*, version 2019/06 (55.197 proteins) [24,26]. To assess the false-positive identification rate, the databases used were reversed “on the fly” during database queries by the software. We used the Progenesis QI for Proteomics software package with Apex3D, Peptide 3D, and Ion Accounting informatics (Waters Corporation) for correct spectral processing and database searching conditions. This software loads the LC-MS data, followed by alignment and peak detection, which creates a list of relevant peptide ions that are analyzed within Peptide Ion Stats by multivariate statistical methods. The processing parameters used were 500 counts for the low-energy threshold and 50.0 counts for the elevated energy threshold. For the processing, all runs in the experiment were automatically aligned and assessed for suitability. For peak picking, eight was used as the maximum ion charge and the sensitivity value was selected as four. In addition, some parameters were considered in the identification of proteins/peptides: (1) digestion by trypsin with at most two missed cleavages; (2) variable modification by oxidation and fixed modification by carbamidomethyl; and (3) a false discovery rate (FDR) of less than 1%. For ion matching, the following were required: two or more ion fragments per peptide, five or more fragments per protein, and one or more peptides per protein. Data were analyzed by one-way analysis of variance (ANOVA) with treatment factor *p*-value < 0.05 compared to the vehicle group, and identifications that were outside these criteria were rejected. The label free protein quantitation was performed using the Hi-N (N = 3) method because the experimental design was defined in 3 groups (1–3 groups). Ratios between the mean values of protein abundances from treated groups, over the mean values of protein abundances from the control group, were calculated for each protein. Proteins which had expression 1.0-fold (log2 fold) increased or decreased in treated groups in comparison to the control group were considered up- or down-regulated.

### 2.6. Functional Correlation Analysis

For interpretation of the functional significance of identified and quantified proteins, gene ontology annotations were performed using Blast2GO Annotation using OmicsBox software. STRING v.11 was used to determine potential interactions between proteins from experimental groups and also between these proteins and other proteins from the STRING database. The Venn diagram was drawn using Draw Venn Diagram Online.

## 3. Results and Discussion

### 3.1. Proteomics Revealed Different Expression Profiles When Macrophages Are Infected with M. smegmatis after Cholesterol Consumption

To study the effects of different nutritional sources provided to *M. smegmatis* when infecting macrophages, proteome proteins were analyzed by Nano-LC-ESI MS/MS. For this, mycobacteria were cultivated under three different conditions and used for infection experiments: (1) M. smegmatis cultivated in complete medium Middlebrook 7H9 broth, which is normally supplemented with glycerol (7H9 + Gly); (2) culture in MM with cholesterol supplementation (MM + Chol); and (3) culture in MM without supplementation (MM). Macrophage proteins were extracted at 12 h after infection with the mycobacterial groups. Macrophages without infection were used as a control group (Figure 1). As a result, a total of 1265 proteins were identified in three independent experiments, among them, 614 were distinct proteins. The mass spectrometry proteomics data were deposited in the ProteomeXchange Consortium via the PRIDE partner repository, with the dataset identifier PXD025783 [27,28,29]. Data demonstrated 90, 58, and 91 exclusive proteins in 7H9 + Gly, MM + Chol, and MM, respectively. The three groups had 276 proteins in common, as shown in the Venn diagram, Figure 2.

All bacterial groups cultured under the different conditions studied (7H9 + Gly, MM + Chol, and MM) modified the proteome of the infected macrophages J774A.1, when compared with uninfected macrophages. In total, 44 proteins were significantly up or down-regulated in macrophages during infection with *M. smegmatis* grown under different nutritional conditions, as shown in Table 1. Specifically, 5 proteins were up-regulated and 2 proteins were down-regulated during infection with *M. smegmatis* grown in 7H9 + Gly. In contrast, 7 proteins were up-regulated and 26 proteins were down-regulated during infection by *M. smegmatis* grown in MM + Chol and 4 proteins were up-regulated and no proteins were down-regulated during infection by *M. smegmatis* grown in MM (Table 1). Thus, after cholesterol consumption, mycobacteria significantly influenced the host macrophage proteome, altering the abundance of 33 proteins, which was significantly more than the alterations induced by the 7H9 + Gly and MM groups. The finding that 26 proteins were down-regulated shows that, after cholesterol consumption, mycobacteria can affect several biological processes and molecular functions in the host cell.

The identified proteins were cataloged into biological processes (Figure 3A), molecular functions (Figure 3B), and cellular components (Figure 3C). According to go annotations from the OmicsBox software, ontology reveals that all proteins altered by the three experimental groups were mainly associated with a cellular process, metabolic process, biological regulation, response to stimuli, and others, as shown in Figure 3A. The ontology also showed that these same proteins are mostly associated with transcription regulator activity, transporter activity, molecular function regulators, structural molecule activity, protein binding, and catalytic activity as the chief molecular function, as shown in Figure 3B. All of these biological processes, molecular function, and cellular components have been previously associated with mycobacteria infection [7,21]. This result suggests a complex interaction between the mycobacteria and the host. 

### 3.2. Common Effects—Immune System and Cytoskeleton

Phagocytosis involves cell surface recognition receptors that transmit signals to various cytoskeletal pathways, to initiate the processes of endocytosis, phagocytosis, vesicular trafficking, and autophagy. In this study, proteins associated with the cytoskeleton, immune response, phagocytosis, autophagy, endocytosis, and vesicular transport were altered in all host proteomes (Appendix A). Among the proteins identified after infection of macrophages with *M. smegmatis* grown in 7H9 + Gly, CORO1C and TUBA1C are down-regulated, and NCOR1, PHLDB2, SQSTM1, and IFITM3 are up-regulated (Appendix A). Only one host protein, IFITM3, was up-regulated by the 7H9 + Gly and MM groups, but not by MM + Chol. Only one cytoskeleton protein, CLASP1, was up-regulated in macrophages infected by *M. smegmatis* grown in MM (Appendix A). In the MM + Chol group, cytoskeleton and immune proteins such as V-ATPase, PARD3, WDR1, MYH7, PFN1, CFL1, and SEPT2 were down-regulated, and KIF15, VIM, and ANXA3 were up-regulated (Appendix A).

#### 3.2.1. Cytoskeleton

The cytoskeleton is essential for phagocytosis in immune cells. Infection of macrophages with *M. smegmatis* grown in 7H9 + Gly down-regulated TUBA1C and CORO1C. These proteins are important for cytoskeleton remodeling, protrusion formation, phagocytosis, and the formation of endocytic vesicles [30,31]. Interference in any of these systems leads to multiple defects in vesicular traffic, formation of vesicles, and endosome fission [32]. In fibroblasts, the absence of coronin-1C affects not only actin filaments, but also microtubules and intermediate vimentin filaments, generating deficiency in cell proliferation and migration [33]. Diaz et al., 2016, also identified CORO1C in exosomes of *M. tuberculosis*-infected cells [21].

In mammals, a function of coronin proteins was initially discovered by studying immune evasion mechanisms used by *M. tuberculosis*. During infection, an important coronin family member, known as TACO or Coronin-1A (CORO-1A), is retained on phagosome macrophages by live mycobacteria to block phagolysosome fusion and thereby prevents the subsequent destruction of the mycobacteria [34,35]. CORO-1A has also been shown to be important for the activation of calcium signaling following mycobacterial entry into macrophages.

The cytoskeleton is involved in all main functions of immune cells related to the response to infection. The architecture of the actin cytoskeleton network is maintained by the coordination of a large number of proteins that regulate the assembly and disassembly of filaments and the contractile force driven by the myosin motor protein. The myosin motor protein can also promote the disassembly of filaments [36]. In the macrophages infected by the MM + Chol group, MYH7, PFN1, CFL1, and WDR1 were all down-regulated. CFL1 and WDR1 are cytoskeleton binding proteins that can work together by inducing the disassembly of actin filaments; their depletion perturbs the actin cytoskeleton [37]. In addition, PFN1 also binds to actin and affects the cytoskeleton structure.

Other cytoskeleton proteins that were up-regulated in macrophages by the MM + Chol *M. smegmatis* group are KIF15, a motor protein of microtubules, ANXA3, and VIM (Vimentine). These last two interact directly and are related to differentiation, migration, phagocytosis, and production of reactive oxygen species [38,39]. VIM is expressed in activated macrophages and responds to pro-inflammatory stimuli [40]. Mahesh et al., 2016, demonstrated that vimentin is up-regulated in macrophages infected with heat-killed *M. tuberculosis* H37Rv and live H37Ra [41]. VIM was also shown to be up-regulated in exosomes of macrophages infected with *M. tuberculosis* [21] and by ManLAM interaction with macrophages [20]. The positive regulation of ANXA3 highlights its role not only as a cytoskeletal protein, but also in blocking the inflammatory process that is inhibited by this protein during infection. ANXA3 blocks phospholipase A2 and consequently prevents the release of arachidonic acid and its modification into prostaglandins and leukotrienes, which are categorized as anti-inflammatory and inflammatory mediators that are important during infection.

#### 3.2.2. Immune System

We analyzed the direct relationship between cytoskeleton proteins, the immune response, and inflammation-related proteins during infection. Previous studies also showed the overexpression of SQSTM1 and IFITM3 in macrophages infected with *M. tuberculosis* [7,20,42]. SQSTM1, up-regulated in the macrophages infected by the 7H9 + Gly group, is one of the best-known substrates for autophagy and participates in selective autophagy; its up-regulation indicates failure in the autophagic process [43,44]. It has been previously shown that phagocytosis is enhanced in autophagy-deficient macrophages, increasing uptake of mycobacteria [43]. IFITM3 was up-regulated in the macrophage infected by the 7H9 + Gly and MM groups; this protein plays a critical role in the structural stability and function of vacuolar ATPase (V-ATPase), a phagosome membrane protein responsible for phagosome acidification and degradation of mycobacteria. IFITM3 restricts mycobacterial growth by mediating endosomal maturation; it acts through interaction with V-ATPase and potentially stabilizes its association with endosomal membranes, increasing the endosomal acidification of cells infected with *M. tuberculosis* [45,46]. 

In contrast, the MM + Chol *M. smegmatis* group induced different responses in macrophages. After cholesterol consumption, *M. smegmatis* induced down-regulation of V-ATPase, indicating a possible destabilization of this phagosome membrane protein. Previous studies have shown that V-ATPase plays important roles during *M. tuberculosis* infection [6,47,48]. In macrophages, *M. tuberculosis* infection inhibits phagosome acidification by the exclusion of V-ATPase and it is critical for *M. tuberculosis* persistence in host macrophages [6]. Shui et al., 2011 showed that v-ATPase was down-regulated by ManLAM (Mtb) interaction with macrophages [20]. Thus, the negative regulation of V-ATPase in this experimental group represents an important finding of our study because this is a key protein in phagosome acidification. These results provide further evidence that *M. smegmatis* induced to cholesterol consumption is able to make changes in the host immune response and contributes to phagosome maturation arrest.

All nutritional conditions offered to *M. smegmatis* affect proteins related to cytoskeleton, immune response, and vesicular traffic of the host during infection. Macrophages infected by *M. smegmatis* grown in MM + Chol demonstrated a greater number of affected proteins than those infected by *M. smegmatis* grown in 7H9 + Gly or MM, indicating that, after cholesterol consumption, mycobacteria can modify the response of the host cell and possibly cause failure in the cytoskeleton, which is important for eliminating mycobacteria. Intracellular pathogens subvert the host cytoskeleton to promote their survival, replication, and dissemination. Actin is a common target of bacterial pathogens, but recent studies have also highlighted the targeting of microtubules, cytoskeletal motors, intermediate filaments, and septins. The study of the cytoskeleton during host–pathogen interactions shed light on key cellular processes such as phagocytosis, autophagy, membrane trafficking, motility, and signal transduction [8], while a recent study showed that *M. tuberculosis* lipids modulate macrophage the actin cytoskeleton [9].

### 3.3. Exclusive Events—Specific Differences in Host Protein Regulation Induced by M. smegmatis Infection

#### 3.3.1. Infection of Macrophages by M. Smegmatis Grown in 7H9 + Gly up-Regulates CABIN1, a Negative Regulator of Calcineurin

Calcium (Ca^2+^) is an important factor in the host–pathogen interaction; it plays a significant role in phagocytosis and is involved in cell division, motility, stress response, signaling, amongst other mechanisms. Previous studies suggest that actin filament and cytoskeleton arrangement alterations require several Ca^2+^-binding proteins/substrates for effective phagocytosis. In the biology of tuberculosis infection, it has been noted that *M. tuberculosis* is able to arrest phagosomal maturation by interfering in Ca^2+^ signaling [49,50].

When macrophages were infected by the 7H9 + Gly group, we identified an exclusive protein that was up-regulated at fold-change 5571 (Appendix A), known as calcineurin binding protein 1 (CABIN1). CABIN1 inhibits calcineurin, which is a Ca^2+^- and calmodulin-dependent ser/thr phosphatase. In the presence of calcium influx, calcineurin inhibits the phagosome-lysosome fusion and also acts an actin-binding protein that adheres to the phagosome membrane and prevents the maturation of phagosome-containing bacilli [49,50]. The up-regulation of CABIN1 may favor the host during 7H9 + Gly *M. smegmatis* infection of macrophages by blocking calcineurin and permiting lysosomal delivery of mycobacteria and phagosome-lysosome fusion. 

The STRING analysis showed that CABIN1 interacts with other proteins related to cell growth, survival, and apoptosis, DNA repair, replication, transcription, and chromosome segregation (Figure 4A). 

#### 3.3.2. Infection of Macrophages with M. Smegmatis Grown in MM Induces up-Regulation of CLASP 1, a Cytoskeleton Protein

The microtubule network in mammalian cells is used by a variety of intracellular pathogens to facilitate their uptake and for the formation, stabilization, and maintenance of their intracellular vacuoles. Macrophage infection with *M. smegmatis* grown only in MM medium differentially regulated a protein at fold-change 5096, known CLIP-associating protein 1 (CLASP1), a microtubule plus-end tracking protein that promotes the stabilization of dynamic microtubules. This protein is involved in cellular adhesion, microtubule organization, immune response and promotes the stabilization of dynamic microtubules via their interaction with actin [51]. Zhao et al. (2013) [51] showed that the silencing of CLASP1 in fibroblasts infected with *Trypanosoma cruzi* reduces internalization of the protozoan and delays fusion of CLASP1-depleted vacuoles with the host lysosomes. Here, for the first time, CLASP1 was related to mycobacterial infection, in the MM group. The up-regulation of CLASP1 may facilitate the internalization of *M. smegmatis* and the phagosome-lysosome fusion in macrophages. These results indicate that *M. smegmatis* cultured in 7H9 + Gly and only in MM may not be able to subvert the immune response of macrophages, which can eliminate the mycobacteria by the positive regulation of two proteins that are related to lysosomal delivery and phagosome-lysosome fusion.

The two differentially-expressed proteins in phagocytes infected with *M. smegmatis* grown in 7H9 + Gly (CABIN1) and MM (CLASP1) were analyzed using STRING-11, with a medium confidence score threshold of 0.4. An interactome network was built for this set of proteins to identify protein-protein interactions and predict functional associations (Figure 4). The STRING analysis of CLASP1 demonstrates its interaction with microtubule cytoskeleton proteins (Figure 4B).

#### 3.3.3. Infection of Macrophages with M. smegmatis Grown in MM + Chol Induces Differential Expression of Proteins of the Immune System and Immunometabolism

In this group, macrophages infected with *M. smegmatis* after cholesterol consumption demonstrated the greatest number of differentially expressed proteins (total of 33). These proteins engage in diverse cellular processes, such as endocytosis and vesicle transport, cytoskeleton rearrangement, ubiquitination pathways, mRNA processing, and immunometabolism. All of these proteins interact with each other, as shown in Figure 5, which suggests a complex interaction between the host and the mycobacteria after cholesterol consumption. Among these proteins, 7 were up-regulated and 26 were down-regulated. 

Among the down-regulated proteins, VDAC2 and HSP90 have been reported in other studies of macrophage activation and infection with *M. tuberculosis* [20,52,53]. Patel et al. (2009) showed that HSP90 has a role in the stabilization of the microtubule cytoskeleton during macrophage activation [52]. Other researchers have reported different functions of host Hsp(s) in controlling bacterial infections. During tuberculosis infection, Hsp(s) exhibit different functions, including toll-like receptor (TLR) activation and immune response induction; they also act as a diagnostic tools and may represent potent vaccine candidates [53].

Interestingly, recent studies have shown that the immune response and metabolic remodeling are interconnected [54]. In response to infection, activation of macrophages makes changes in the bioenergetic pathway from oxidative phosphorylation to glycolysis, and *M. tuberculosis* infection perturbs this pathway to facilitate its survival and persistence [54,55,56,57,58]. Cumming et al. (2018) [58] demonstrated that *M. tuberculosis* induces a quiescent energy phenotype in human monocyte-derived macrophages and decelerated flux through glycolysis and the TCA cycle. Furthermore, *M. tuberculosis* reduced mitochondrial dependency on glucose and increased mitochondrial dependency on fatty acids. A proteomic analysis by Li et al. (2017) [7] demonstrated that, during *M. tuberculosis* infection of THP-1, the most modulated proteins were mainly implicated in metabolic processes important in TB pathogenesis and transmission.

This experimental group demonstrated interference in proteins associated with host metabolism. A total of 11 proteins were differentially regulated; among them, aldo-keto reductase (AKR1A1), aldehyde dehydrogenase (ALDH16A1), aldose reductase (AKR1B8), phosphoglycerate mutase 1 (PGAM1), and transketolase (TKT) were down-regulated and other mitochondrial proteins such dihydrolipoyllysine-residue succinyltransferase (DLST) and fumarate hydratase (FH) were up-regulated. Multiple studies have shown that the perturbation of key metabolic enzymes can affect the immune functions of macrophages [4]. Some metabolic enzymes such as PGAM1, TKT, and ALDH16A1 have also been reported to be associate with the cytoskeleton [52,59].

Microorganisms can modulate the metabolic status of macrophages using virulence factors such as cell wall constituents. Our findings indicate that, after cholesterol consumption, *M. smegmatis* may induce similar responses to *M. tuberculosis* in host proteins, down-regulating glycolytic and redox enzymes, and upregulating mitochondrial enzymes. Among the 33 proteins differentially regulated in macrophages by *M. smegmatis* cultured in MM + Chol, 20 proteins interacted with each other, as shown in Figure 5. ANXA3, a member of the calcium-dependent phospholipid-binding protein family that inhibits the function of phospholipase A2, interacted with VIM which integrates the cytoskeleton. ANXA3 also interacts with WDR1, an actin-binding protein that interacts with other cytoskeleton proteins such as PFN1 and CFL1, demonstrating the direct relationship between the cytoskeleton and the immune response. Other proteins, for instance HSP90, TPT1, SKP1A, PSMD1, PSMD2, and PPIA, are related to protein folding, stabilization, and degradation. PRPF19, HNRNPL, and SYNCRIP are related to mRNA processing. Six proteins, DLST, FH1, PGAM1, TKT, AKR1A1, and AKR1B8, which are involved in immunometabolism, show interactions with each other.

## 4. Conclusions

The alterations of different biological processes in macrophages after infection with *M. smegmatis* grown in different nutritional conditions helps us to understand that, when bacilli use the carbon source available in the environment, this modifies their metabolism and induces different macrophage responses. In particular, cholesterol consumption by *M. smegmatis* grown in MM causes several changes in cell wall bioactive components. We hypothesized that cholesterol-modified bioactive molecules may be interacting with macrophages to down-regulate many proteins involved in key biological processes within the cell. Thus, *M. smegmatis* could be modulating the immune response to remain viable within the host. 

During macrophage infection with *M. smegmatis* cultured in 7H9 + Gly complete medium, only 7 proteins related to the cytoskeleton, vesicular traffic, and immune response were differentially regulated. In this group, most proteins presented up-regulation (e.g., SQSTM1, IFITM3, NCOR1, PHLDB2, and CABIN1), indicating enhancement of phagocytosis, stabilization of the V-ATPase phagosome protein, endosomal acidification, and favoring of phagosome-lysosome fusion. Similarly, macrophages infected with *M. smegmatis* grown in MM without any supplementation induced only up-regulation of proteins such as IFITM3 and CLASP1, indicating that macrophages are able to eliminate mycobacteria by the positive regulation of proteins related to the stabilization of the phagosome and dynamic microtubules, allowing lysosomal delivery and phagosome-lysosome fusion.

In contrast, the most interesting results in our study were observed for macrophages infected with *M. smegmatis* that were cultured in MM + Chol. Here, mycobacterial infection induced greater differential expression of proteins. Of the 33 differentially expressed proteins, 26 were down-regulated, including V-ATPase, VDAC2, HSP90, TPT1, PPIA, PARD3, MYH7, PFN1, WDR1, CFL1, SEPT2, METAP2, APRT, LARS, TKT, PGAM1, AKR1A1, ALDH16A1, and AKR1B8. These proteins are related to several biological processes such as cytoskeleton remodeling, the ubiquitination pathway, immune response, mRNA processing, and immunometabolism. Our data reveal that, after cholesterol consumption, the mycobacteria interact with macrophages in a more complex way, down-regulating diverse proteins that are important for cytoskeleton remodeling, an essential biological process for phagocytosis and endocytosis. In addition, negative regulation of V-ATPase indicates interference in phagosome acidification, as occurs during *M. tuberculosis* infection. Another important finding of this study was the down-regulation of immunometabolism proteins that have recently been found to be important for the host–pathogen interaction. Recent studies have reported that *M. tuberculosis* causes metabolic remodeling of immune cells to facilitate its survival and to persist in the host [55]. Here, the saprophytic *M. smegmatis*, following cholesterol consumption, is able to perturb macrophage proteins involved in metabolism, similarly to *M. tuberculosis*. Santos et al., 2019, showed that *M. smegmatis*, grown in MM medium with cholesterol supplementation, demonstrates changes in its bioactive cell wall components, cell surface hydrophobicity, acquires H_2_O_2_ resistance, and presents granuloma like-structure formation [16]. Thus, these results indicate that the modification of bioactive cell wall molecules of *M. smegmatis* by cholesterol consumption may modulate the immune response of macrophages through down-regulation of key proteins related to the biological processes of the immune response, immunometabolism and cytoskeleton remodeling. As such, these bacterial may be able to subvert the defense mechanisms of macrophages to facilitate their survival inside the host. 

In conclusion, our study provides the first proteomic analysis that compares a large number of cellular proteins that are differentially regulated in macrophages by *M. smegmatis*, after inducing cholesterol consumption or not. The dynamic responses to infection caused by these mycobacteria are also characterized. This information will be important for understanding how mycobacteria use the carbon source available in the environment to manipulate the metabolism and defense mechanisms of the host macrophage. A more comprehensive understanding of host–pathogen interaction, cellular metabolism, and signaling networks may provide a novel avenue to improve disease therapeutics, vaccines, and biomarkers for *M. tuberculosis* infection. Furthermore, understanding how microbial metabolism interacts with macrophage metabolism and how this influences the control or progression of infection may shape future investigations.

## Figures and Tables

**Figure 1 pathogens-10-00662-f001:**
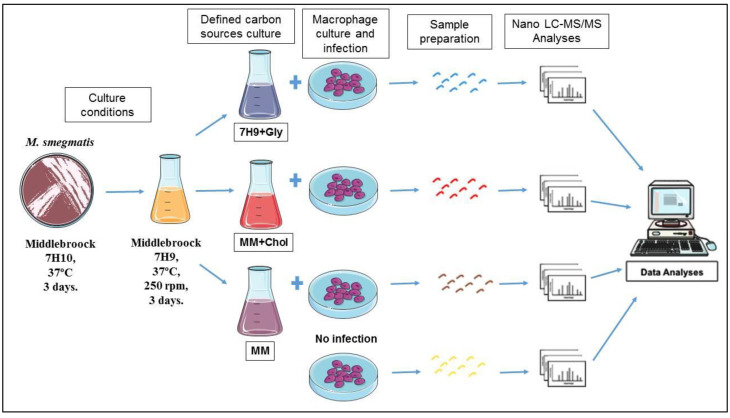
Experimental design. The macrophage cell line J774A.1 was cultivated and infected with *M. smegmatis* grown in different nutritional conditions. Proteins were extracted and analyzed by Nano-LC-ESI MS/MS. As a control, uninfected macrophages were used.

**Figure 2 pathogens-10-00662-f002:**
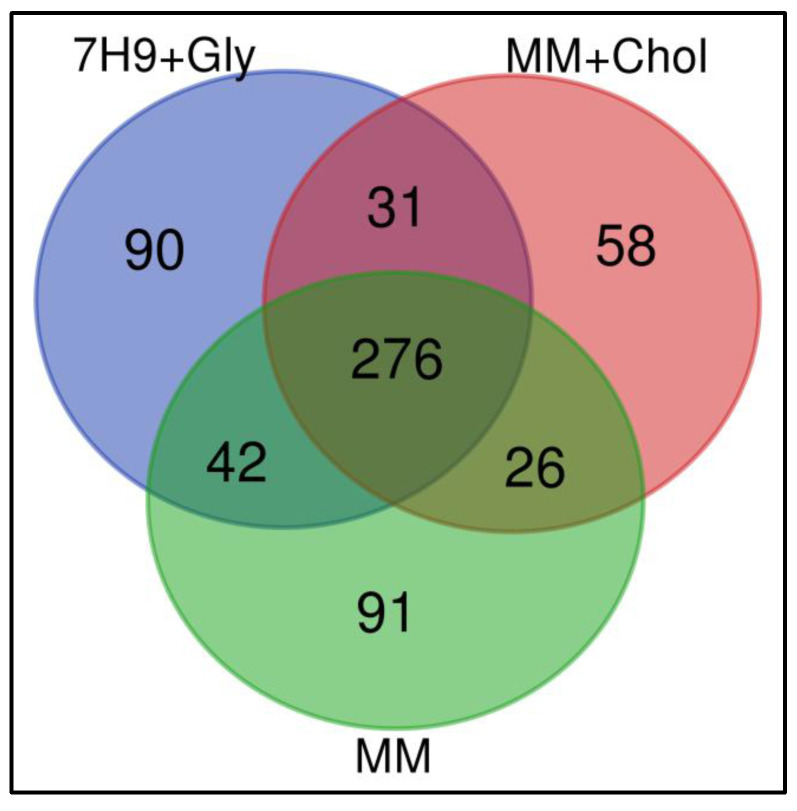
Qualitative analyses of all identified proteins. Venn diagram shows the overlap in the numbers of protein identifications between macrophages infected with *M. smegmatis* after their culture in Middlebrook 7H9 broth (7H9 + Gly), minimal medium with cholesterol supplementation (MM + Chol), and minimal medium without supplementation (MM).

**Figure 3 pathogens-10-00662-f003:**
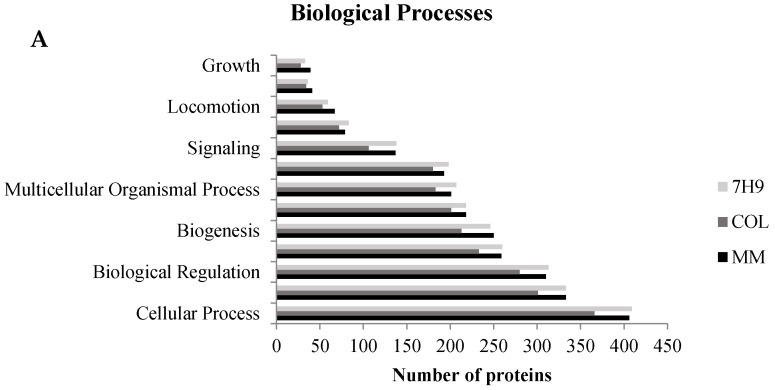
Gene ontology analysis of all identified proteins. Macrophage proteins were analyzed according to their biological process (**A**), molecular function (**B**), and cellular component (**C**). This classification was produced based on an analysis using the Blast2GO Annotation through OmicsBox software.

**Figure 4 pathogens-10-00662-f004:**
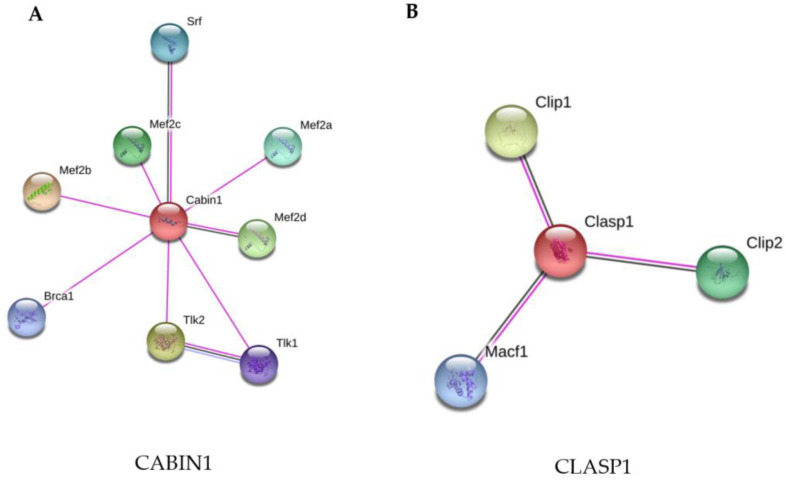
STRING analysis of the CABIN1 and CLASP1 proteins. (**A**) CABIN1 was up-regulated in macrophages infected with *M. smegmatis* after their culture in Middlebrook 7H9 broth (7H9 + Gly). (**B**) CLASP1 was up-regulated in macrophages infected with *M. smegmatis* after their culture in minimal medium without supplementation (MM). The proteins were grouped using the STRING software version 11.0. Line colors: known interactions, purple line (determined experimentally); predicted interactions, dark blue (gene co-occurrence); other, black (co-expression).

**Figure 5 pathogens-10-00662-f005:**
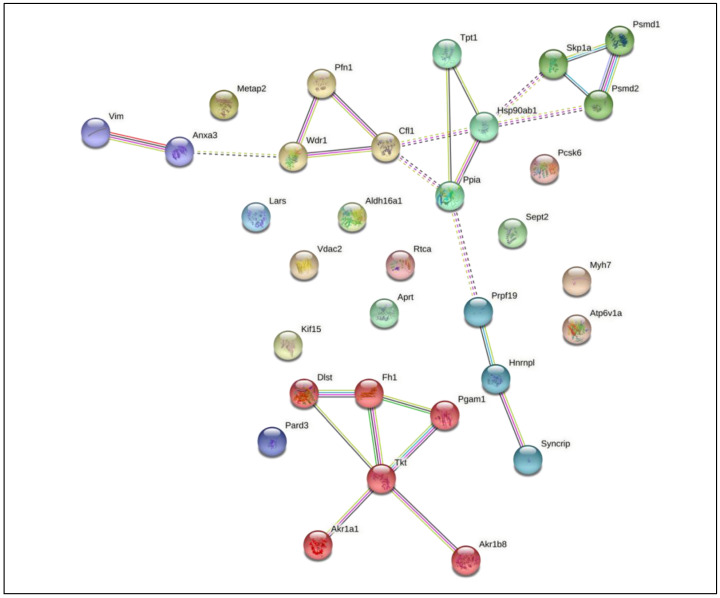
STRING analysis of differentially expressed proteins in macrophages infected with *Mycobacterium smegmatis* after cholesterol consumption (MM + Chol) revealed 20 interaction partners. The proteins were grouped using the STRING software version 11.0. Line colors: known interactions, purple line (determined experimentally); predicted interactions, dark blue (gene co-occurrence); other, black (co-expression).

**Table 1 pathogens-10-00662-t001:** Quantification of identified, quantified, exclusive, and differentially regulated proteins in proteomic analysis.

Experimental Groups	Identified Proteins	Quantified Proteins	Exclusive Proteins	Regulated Proteins	
				Up-	Down-
7H9 + Gly	439	119	90	5	2
MM + Chol	391	139	58	7	26
MM	435	90	91	4	0
Total	1265			44	

## Data Availability

The mass spectrometry proteomics data have been deposited at the ProteomeXchange Consortium via the PRIDE partner repository with the dataset identifier PXD025783.

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
