# Peer review of "Culture of Mycobacterium smegmatis in Different Carbon Sources to Induce In Vitro Cholesterol Consumption Leads to Alterations in the Host Cells after Infection: A Macrophage Proteomics Analysis"

_pathogens, 2021, doi:10.3390/pathogens10060662_

Round 1
Reviewer 1 Report
The manuscript ‘Mycobacterium smegmatis cultured in different carbon sources to induce in vitro cholesterol consumption reveals alterations in the host cells after infection: a macrophage proteomics analysis’ by Jaqueline Batista de Lima and co-workers continues a series of topical works to establish a detailed mechanism of host-pathogen interaction during mycobacterial infections. Remarkably, in this work the authors have discovered that after cholesterol consumption non-pathogenic M. smegmatis bacilli were able to perturb macrophages proteins during infection in the same way as pathogenic M. tuberculosis. In particular, when grown on cholesterol, M. smegmatis down-regulates diverse proteins of cytoskeleton, immunometabolilic proteins and interfere essential biologic processes of phagocytosis and endocytosis like M. tuberculosis, which is known to cause metabolic remodeling of immune cells to facilitate bacterial survival and persistence in the host. However, having read this manuscript, I have some critical comments.
1) Text needs careful editing. For instance, lines 83-85: ‘… using it as a carbon source and biosynthetic precursors that are needed to produce the virulence associated [2], [13]–[15]’. Some word(s) are missing here?
2) Lack of explanation of abbreviation. For instance, see line 65 “cell wall components like PIMs, LM, LAM, TDM, GPL…”
3) Figure 2 duplicates heavily Table 1
4) Table 1 is absolutely misleading and needs considerable improvement. Line ‘Total 1265’ in the bottom is absolutely confusing.
Author Response
1) Text needs careful editing. For instance, lines 83-85: ‘… using it as a carbon source and biosynthetic precursors that are needed to produce the virulence associated [2], [13]–[15]’. Some word(s) are missing here?
Response: The sentence was rewritten: “where the bacterium uses it as a source of carbon and biosynthetic precursors that are needed to produce virulence-associated molecules”.
2) Lack of explanation of abbreviation. For instance, see line 65 “cell wall components like PIMs, LM, LAM, TDM, GPL…”
Response: The explanations of abbreviations were added: phosphatidylinositol mannosides (PIMs), lipomannan (LM), lipoarabinomannan(LAM), trehalose dimycolate (TDM), glycopeptidolipid (GPL).
3) Figure 2 duplicates heavily Table 1.
Response: The table 1 was rewritten because some numbers were missing. We showed the data in this table 1 to improve with more details the data showed in figure 2.
4) Table 1 is absolutely misleading and needs considerable improvement. Line ‘Total 1265’ in the bottom is absolutely confusing.
Response: The table 1 was rewritten because some numbers were missing.
Reviewer 2 Report
The study of Batista de Lima et al., describes the effect on macrophages proteome infected with M. smegmatis cells grown on different media. The authors show interestingly that M. smegmatis bacteria fed with cholesterol (known to induce cell wall composition modifications) triggers regulation of several important proteins belonging to different pathways in macrophages thus demonstrating a “global and complex effect” on the host albeit the number of proteins affected is somehow limited.
The study is overall of interest and well-conducted. The paper would be suitable for publication after minor modifications.
Points to address:
-Line 245: the authors stated “44 proteins were significantly up or down-regulated in macrophages during infection” what are the criteria here in this study to consider a significant difference of expression?
-In this study, the mycobacteria are “artificially” fed with cholesterol, as this is not mentioned in the discussion could the authors may relate/discuss this to what happens in a “real” infection. Do we know how much cholesterol is taken up by mycobacteria? This is just to appreciate how physiologically relevant that study is.
-Remark on reference: the global TB report 2020 is available so please update
-Line 47: The authors state “One of the major obstacles to control TB is our lack of understanding of the host-pathogen interaction”. I think this statement is a bit strong we know quite a lot but not everything for sure so please rephrase and moderate this claim.
-Other points mainly writing issue
-There is no abbreviation definition for PDIM, LAM, LAM...and so on
that are quite some spelling/grammatical mistakes or words forgotten so please read again carefully, I guess these points will also be addressed during the journal proofs.
-for example, line 74 “is recruited...’, line 90 “they also that” show is missing ….and many more.
-line 127: “0.05% (v/v) tyloxapol on ethanol” on ethanol I did not understand this ??
-line 129: “were diluted to O.D.600 0.05 ml in fresh 7H9” some words missing here either OD value or ml value.
-line 159 : “the samples were diluted in 10X 20 mM ammonium bi-carbonate pH 7.5” is 20mM ammonium bi-carbonate the 10x stock or the authors used 10x 20mM i.e. 200mM please rephrase.
Author Response
-Line 245: the authors stated “44 proteins were significantly up or down-regulated in macrophages during infection” what are the criteria here in this study to consider a significant difference of expression?
Response: To explain it in greater detail we added the follow sentence at Data Processing and Database Searches of Materials and Methods: "Ratios between the mean values of protein abundances from treated groups, over the mean values of protein abundances from the control group, were calculated for each protein. Proteins which had expression 1.0-fold (log2 fold) increased or decreased in treated groups in comparison to control group were considered up- or downregulated".
-In this study, the mycobacteria are “artificially” fed with cholesterol, as this is not mentioned in the discussion could the authors may relate/discuss this to what happens in a “real” infection. Do we know how much cholesterol is taken up by mycobacteria? This is just to appreciate how physiologically relevant that study is.
Response: We rewrite the paragraph, when we describe the role of in vitro cholesterol consumption and its association with the real infection. We do not know the concentration of cholesterol that is take up by mycobacteria, but we improve the informations to show what we have about it: “The authors demonstrated that cholesterol availability in MM doubled the amount of intracellular cholesterol, only when glycerol is lacking, showing it to be important for the maintenance of small lipids such as PIMs and phospholipids on the mycobacterial surface”. It is related to our previously publication of SANTOS et al., (2019).
-Remark on reference: the global TB report 2020 is available so please update
Response: It was done. We replaced the wrong reference with the actual global TB report 2020.
-Line 47: The authors state “One of the major obstacles to control TB is our lack of understanding of the host-pathogen interaction”. I think this statement is a bit strong we know quite a lot but not everything for sure so please rephrase and moderate this claim.
Response: We rephrased the phrase: “The improvement of knowledge regarding the host-pathogen interaction is one manner of overcoming obstacles to control TB.”
-Other points mainly writing issue.
Response: We did another review of the manuscript.
-There is no abbreviation definition for PDIM, LAM, LAM...and so on.
Response: The explanations of abbreviations were added in the new version of manuscript.
that are quite some spelling/grammatical mistakes or words forgotten so please read again carefully, I guess these points will also be addressed during the journal proofs.
Response: We did another review of the manuscript.
-for example, line 74 “is recruited...’, line 90 “they also that” show is missing ….and many more.
Response: We did another review of the manuscript.
-line 127: “0.05% (v/v) tyloxapol on ethanol” on ethanol I did not understand this ??
Response: We rewrite it: "0.05% (v/v) tyloxapol".
-line 129: “were diluted to O.D.600 0.05 ml in fresh 7H9” some words missing here either OD value or ml value.
Response: The sentence was rewritten: “The bacterial culture in the stationary phase of growth was diluted to get a new 20 mL culture with O.D.600 0.05 in fresh Middlebrook 7H9 broth, as before, or in minimal media (MM)”
-line 159 : “the samples were diluted in 10X 20 mM ammonium bi-carbonate pH 7.5” is 20mM ammonium bi-carbonate the 10x stock or the authors used 10x 20mM i.e. 200mM please rephrase.
Response: The sentence was rewritten: "the samples were 10-folds diluted in 20 mM ammonium bicarbonate"